# TRPML1-Induced Lysosomal Ca^2+^ Signals Activate AQP2 Translocation and Water Flux in Renal Collecting Duct Cells

**DOI:** 10.3390/ijms24021647

**Published:** 2023-01-13

**Authors:** Simona Ida Scorza, Serena Milano, Ilenia Saponara, Maira Certini, Roberta De Zio, Maria Grazia Mola, Giuseppe Procino, Monica Carmosino, Francesco Moccia, Maria Svelto, Andrea Gerbino

**Affiliations:** 1Department of Biosciences, Biotechnologies and Environment, University of Bari Aldo Moro, 70125 Bari, Italy; 2Department of Sciences, University of Basilicata, 85100 Potenza, Italy; 3Department of Biology and Biotechnology Lazzaro Spallanzani, University of Pavia, 27100 Pavia, Italy

**Keywords:** aquaporin 2, lysosome, TRP channels, Ca^2+^ signaling, nephron, collecting duct, kidney

## Abstract

Lysosomes are acidic Ca^2+^ storage organelles that actively generate local Ca^2+^ signaling events to regulate a plethora of cell functions. Here, we characterized lysosomal Ca^2+^ signals in mouse renal collecting duct (CD) cells and we assessed their putative role in aquaporin 2 (AQP2)-dependent water reabsorption. Bafilomycin A1 and ML-SA1 triggered similar Ca^2+^ oscillations, in the absence of extracellular Ca^2+^, by alkalizing the acidic lysosomal pH or activating the lysosomal cation channel mucolipin 1 (TRPML1), respectively. TRPML1-dependent Ca^2+^ signals were blocked either pharmacologically or by lysosomes’ osmotic permeabilization, thus indicating these organelles as primary sources of Ca^2+^ release. Lysosome-induced Ca^2+^ oscillations were sustained by endoplasmic reticulum (ER) Ca^2+^ content, while bafilomycin A1 and ML-SA1 did not directly interfere with ER Ca^2+^ homeostasis per se. TRPML1 activation strongly increased AQP2 apical expression and depolymerized the actin cytoskeleton, thereby boosting water flux in response to an hypoosmotic stimulus. These effects were strictly dependent on the activation of the Ca^2+^/calcineurin pathway. Conversely, bafilomycin A1 led to perinuclear accumulation of AQP2 vesicles without affecting water permeability. Overall, lysosomal Ca^2+^ signaling events can be differently decoded to modulate Ca^2+^-dependent cellular functions related to the dock/fusion of AQP2-transporting vesicles in principal cells of the CD.

## 1. Introduction

Lysosomes play key roles in the molecular machinery that enables normal renal physiology. One critical function of lysosomes is the catabolism of low-molecular-weight proteins that freely cross the glomerular filter into the lumen of the nephron. These small proteins are immediately endocytosed by the cells of the proximal tubules with a receptor-dependent mechanism and targeted to the lysosomes for degradation, thus avoiding their waste with urine excretion [1]. A second pivotal role for lysosomes in the kidney is related to the regulation of water and electrolyte homeostasis. Water reabsorption in the collecting duct relies on the vasopressin-dependent exocytosis of intracellular vesicles containing the water channel aquaporin 2 (AQP2) at the luminal surface of principal cells, thereby increasing water permeability [2]. Then, apical AQP2 is constitutively internalized by the endosomal system and is either recycled or targeted for lysosomal degradation to finely tune the amount of AQP2 available for water transport [3]. The antidiuretic hormone vasopressin is crucial to regulate AQP2 retention on the apical membrane [2]. Activation of the vasopressin type 2 receptor (V2R) leads to a cytosolic increase in cAMP that stimulates protein kinase A (PKA) to phosphorylate AQP2. This signaling pathway leads to apical fusion and longer retention of AQP2-containing intracellular vesicles on the plasma membrane, thereby increasing the water permeability of the apical membrane [2].

The importance of lysosomes in renal physiology is also highlighted by the evidence that diseases involving abnormal lysosomal functionality, such as lysosomal storage disorders (LSD), often manifest severe renal phenotypes [4]. For instance, Fabry’s disease [5,6] and cystinosis [7,8] involve the accumulation of catabolites within kidney lysosomes and are usually characterized by inflammation and tubulointerstitial fibrosis, which are common features of end-stage renal diseases [4]. In addition, it has been recently shown that abnormal expression of specific lysosomal proteins might result in defective water reabsorption. In mice, the genetic ablation of lysosomal Limp1, also known as CD63, results in abnormal intracellular lamellar inclusions in the principal cells, along with polyuria and reduced urine osmolality [9]. In this context, polyuria relies upon the formation of intracellular inclusion bodies that sterically impair AQP2 trafficking at the apical surface of the collecting duct cells. However, the precise connection between lysosomes function and vasopressin-induced water reabsorption at the collecting duct is still foggy.

This study was inspired by recent discoveries that led to a re-evaluation of the mere digestive functions of this organelle, which is now regarded as a sophisticated regulatory hub that integrates multiple signals to finely tune different cell functions, such as nutrient sensing, intracellular signaling, vesicle trafficking and cellular growth [10]. The multifaceted role of lysosomes is granted by the presence of about 25 transmembrane proteins and a lysosomal matrix that actively stores the highest amount of free Ca^2+^ (0.5 mM) into cellular organelles with the sole exception of the endoplasmic reticulum (ER) [11]. The accumulation of such a large amount of Ca^2+^ within lysosomes still puzzles scientists. On one hand, the lysosomal matrix can be refilled with Ca^2+^ in a pH-dependent manner owing to the presence of a putative H+− Ca^2+^ exchanger [11,12,13]. On the other hand, Ca^2+^ uptake through the extracellular space via endocytosis or store-operated Ca^2+^ entry seems to be the most reasonable scenario [14]. However, ER Ca^2+^ could also be redirected towards the lysosomal matrix via inositol-1,4,5-trisphosphate receptors (InsP3Rs) [15]. As described for the ER, two distinct second messengers, such as NAADP and PI(3,5)Ps, can rapidly mobilize lysosomal Ca^2+^ by gating two pore channels TPC1 and TPC2 [16] and Transient Receptor Potential Mucolipin 1 (TRPML1) [17], respectively. Mobilization of lysosomal Ca^2+^ stores via TPC1 and TPC2 has long been known to evoke ER-dependent Ca^2+^ release and, therefore, shape intracellular Ca^2+^ signals in different cell types [18,19,20,21,22]. Lysosome–ER interplay may convert local, peri-lysosomal Ca^2+^ release events into global Ca^2+^ elevations to finely tune distinct physiological processes, such as membrane fusion and vesicular trafficking [23] and proliferation [24]. Conversely, only scarce evidence has been presented in support of ER Ca^2+^ release events triggered by local lysosomal Ca^2+^ mobilization via TRPML1 [25,26]. Of note, impaired lysosomal Ca^2+^ signaling often associates with life-threatening diseases, such as cancer [17], viral infections [27], hypertension and arrhythmia [28] and lysosomal storage disorders [29,30,31], underlying the biomedical need to precisely delineate lysosomal Ca^2+^ signaling in full.

Therefore, the aim of this work is to provide the first evaluation of lysosomal Ca^2+^ signaling in renal collecting duct cells by either blocking the vacuolar H-ATPase (V-ATPase) with bafilomycin A1 [32] to deplete the lysosomal Ca^2+^ pool or activating TRPML1 with the synthetic agonist ML-SA1 [33]. We found that both lysosomal agonists induced robust and long-lasting cytosolic Ca^2+^ oscillations sustained by ER Ca^2+^ release through Ins3PRs. In addition, we aimed at understanding whether lysosomal Ca^2+^ release drives AQP2-dependent water reabsorption. Of note, we showed that ML-SA1 and bafilomycin A1 differentially affected AQP2 translocation to the apical membrane and actin polymerization in the cytosol, and that only ML-SA1 induced maximal water reabsorption in collecting duct cells. In our experiments, even though ML-SA1 was strikingly able to increase water permeability to an extent similar to that induced by submaximal doses of the cAMP increasing agents forskolin and IBMX, TRPML1 activation did not affect the cAMP/PKA pathway. Conversely, TRPML1-dependent AQP2 translocation and actin depolymerization were inhibited by buffering intracellular Ca^2+^ changes with BAPTA or blocking the Ca^2+^-dependent phosphatase calcineurin (CaN) with cyclosporine A. Finally, ex vivo stimulation of freshly isolated mouse kidney slices with the same dose of ML-SA1 confirmed the existence of the same modulatory action of TRPML1 activation on AQP2 accumulation even in a more complex physiological system.

## 2. Results

### 2.1. Endo-Lysosomal System Distribution and Ca^2+^ Homeostasis Characterization of M1 Cells

M1 mouse kidney cortical collecting duct (CD) cells are densely packed with acidic endosomes/lysosomes, as evident by the positive loading with the weak base of the Lysosome Staining Kit (LSK) (Figure 1A). Lysosome immunolocalization with LAMP-1 revealed well-resolved vesicular structures throughout the cell (Figure 1B). These cells have been extensively characterized by our group in terms of intracellular signaling events [34,35,36]. When loaded with the Ca^2+^-sensitive fluorophore, Fura-2-AM, each M1 cell showed asynchronous Ca^2+^ oscillation in terms of frequency and amplitude (Figure 1C).

This behavior is likely related to the absence of gap junctions and thus defective intracellular Ca^2+^ coupling, as reported for inner medullary collecting duct cells [37]. In addition, Ca^2+^ oscillations ceased after the removal of extracellular Ca^2+^, indicating a high resting membrane permeability for Ca^2+^. In agreement with this observation, the prompt decrease in fluorescence observed at the isosbestic point for Fura-2 (360 nm) when extracellular Ca^2+^ was replaced with Mn^2+^ hints at the presence of a constitutive Ca^2+^ entry pathway (Figure 1D), as discussed in [38]. This high permeability reflects the expression of several TRP channels at the plasma membrane of these cells, as previously reported [34].

### 2.2. Evaluation of Lysosomal Ca^2+^ Signaling Events in M1 Cells

To dissect lysosomal Ca^2+^ signaling events in M1 cells, we induced lysosomal Ca^2+^ release by alkalizing the intravesicular pH that maintains a high Ca^2+^ level within the lysosomal matrix [11,21,22]. In this context, 200 µM Gly-Phe b-naphthylamide (GPN) has been used for more than 20 years to induce and study lysosomal Ca^2+^ signaling events, also in renal cells [39]. This weak hydrophobic base accumulates specifically within lysosomes and its degradation produces free amino acids that osmotically permeabilize the lysosomal membrane [40]. In accord, 200 µM GPN caused a fast dissipation of the fluorescence signal of the LSK within 2 min in M1 cells (Figure 2A, LSK fluorescence intensity/min: −0.24 ± 0.03, *n* = 89 cells). Parallel measurements of cytosolic Ca^2+^ concentration in cells loaded with Fura-2 showed that acute addition of 200 µM GPN induced, in about 70% of the cells, asynchronous Ca^2+^ transients that were qualitatively compared in each experiment with 100 μM ATP-induced Ca^2+^ increases (Figure 2B). Note that in line with its action, GPN-induced Ca^2+^ responses do not resemble those evoked by the activation of Gq-coupled purinergic receptors. In addition, GPN perfusion reversibly increased the cytosolic pH to a lower extent as compared to the addition of NH_4_Cl, as shown in BCECF-loaded M1 cells (Figure 2C, ΔRatio BCECF: GPN 0.21 ± 0.003 vs. NH_4_Cl 0.71 ± 0.007, *p* < 0.0001). This effect has been previously associated with the fact that GPN is a weak base that can potentially alkalize the cytosol, thereby inducing ER-dependent Ca^2+^ changes unrelated to lysosomal membrane permeabilization and lysosomal Ca^2+^ release [41]. However, the Sandip Patel group has subsequently shown that GPN-induced alkalization is actually dissociated from changes in cytosolic Ca^2+^ and that the Ca^2+^ signaling events evoked upon activation of lysosomal Ca^2+^ channels were specifically inhibited by GPN via lysosomal permeabilization [42].

Nonetheless, to investigate lysosomal Ca^2+^ signaling events in M1 cells, we decided to avoid the use of GPN and, therefore, selected two alternative strategies to induce lysosomal Ca^2+^ release: (i) we alkalized the intravesicular pH using a specific blocker of the V-ATPase, bafilomycin A1 [11], and (ii) we used the synthetic agonist ML-SA1 to activate the ubiquitous Ca^2+^-permeable TRPML1 channel, which is localized in the late endosomal and lysosomal compartment [42]. As shown in Appendix A, Western blot and immunofluorescence experiments clearly indicate that TRPML1 is expressed in M1 cells (and in a mouse kidney lysate), where it shows a vesicular distribution throughout the cell. As shown in Figure 3A, we confirmed that bafilomycin A1 slowly reduced the accumulation of the LSK within the endo-lysosomal compartment when compared with GPN (LSK fluorescence intensity/min: BA1 −0.06 ± 0.002 vs. GPN −0.24 ± 0.03, *p* = 0.0001), but it did not change the cytosolic pH (Figure 3B).

On the other hand, ML-SA1 did not change either the lysosomal accumulation of LSK (Figure 3C) or the cytosolic pH (Figure 3D). These experiments corroborate the use of bafilomycin A1 and ML-SA1 as lysosomal Ca^2+^ releasing agents that do not affect cytosolic pH. In the presence of extracellular Ca^2+^, the Ca^2+^ responses induced by bafilomycin A1 and ML-SA1 were different. Bafilomycin A1 (100 nM, Figure 3E) induced, in 23% of the cells, long-lasting Ca^2+^ oscillations that persisted after its removal. On the other hand, 100 µM ML-SA1 induced, in 80% of the cells, Ca^2+^ responses that were spatio-temporally grouped in two patterns, namely rapid and persistent Ca^2+^ increases, which were likely involving TRPML1 putatively recycled at the plasma membrane (Figure 3F, blue trace, 75% of the responsive cells), and Ca^2+^ oscillations similar to those observed with bafilomycin A1 that disappeared upon ML-SA1 removal (Figure 3F, red traces, 25% of the responsive cells). Thus, to isolate lysosomal Ca^2+^ release events, we challenged the cells with bafilomycin A1 and ML-SA1 in the absence of extracellular Ca^2+^. Under this experimental condition, perfusion with either bafilomycin A1 (Figure 3G) or ML-SA1 (Figure 3H) elicited persistent Ca^2+^ oscillations that exclusively reflect intracellular Ca^2+^ release events from endogenous stores. Removal of extracellular Ca^2+^ changed the number of cells sensitive to both lysosomal agents. Stimulation with bafilomycin A1 in the absence of extracellular Ca^2+^ induced Ca^2+^ oscillations in a larger number of cells (in %: Ca^2+^ 23.86 ± 3.84 vs. free Ca^2+^ 73.51 ± 4.51, *p* = 0.0093). Conversely, perfusion with ML-SA1 in free Ca^2+^ induced Ca^2+^ oscillation in a reduced number of cells (in %: Ca^2+^ 80.21 ± 0.32 vs. free Ca^2+^ 51.57 ± 2.93, *p* = 0.0002). The apparent discrepancy between these findings could be due to the requirement of spontaneous Ca^2+^ oscillations for extracellular Ca^2+^ (Figure 1C). If lysosomal Ca^2+^ stores support unsolicited intracellular Ca^2+^ spikes through TPCs [24], bafilomycin A1 could fail to evoke detectable Ca^2+^ signals as its target organelle is already releasing Ca^2+^. Conversely, the removal of extracellular Ca^2+^ results in the interruption of the ongoing Ca^2+^ activity, thereby unmasking bafilomycin A1-evoked intracellular Ca^2+^ oscillations. On the other hand, the robust mobilization of lysosomal Ca^2+^ induced in the presence of extracellular Ca^2+^ by the direct stimulation of TRPML1 suggests that ML-SA1 targets a lysosomal store that is silent during spontaneous Ca^2+^ oscillations and is partially depleted with Ca^2+^ in the absence of extracellular Ca^2+^ [43].

To further investigate if Ca^2+^ oscillations rely specifically on lysosomal Ca^2+^, we examined the effect of bafilomycin A1 and ML-SA1 after GPN-induced osmotic rupture of lysosomes. As shown in Figure 4A,B a 30 min pretreatment with 200 µM GPN was sufficient to significantly blunt the percentage of cells sensitive to bafilomycin A1 (in %: bafilomycin A1 73.51 ± 4.51 vs. GPN + bafilomycin A1 5.15 ± 4.31, *p* = 0.0124) and ML-SA1 (in %: ML-SA1 51.57 ± 2.93 vs. GPN + ML-SA1 2.19 ± 2.33, *p* = 0.0155) without affecting the response induced by ATP (data not shown). In addition, pretreatment with ML-SI1, a specific antagonist of TRPML1 that did not affect Ca^2+^ homeostasis per se [44], significantly reduced the cells responding to ML-SA1 with Ca^2+^ oscillations (Figure 4C, in %: ML-SA1 51.57 ± 2.93 vs. ML-SI1 + ML-SA1 3.74 ± 0.73, *p* = 0.0062). This finding clearly confirms the specificity of ML-SA1 as a synthetic agonist for TRPML1. 

Taken together, these findings show for the first time that lysosomal Ca^2+^-signaling events mediated by the activation of TRPML1 are reported in collecting duct cells. 

### 2.3. Evaluation of Lysosome and ER Ca^2+^ Interplay

In other cell types, lysosomal Ca^2+^ release may be sustained by either lysosomal Ca^2+^ uptake from the ER [15,39,45] or by ER-dependent Ca^2+^ mobilization through the “Ca^2+^-induced Ca^2+^ release” (CICR) mechanism, which results in the appearance of regenerative Ca^2+^ waves [13,21,22]. The persistence of bafilomycin A1-induced Ca^2+^ oscillation after agonist removal suggests that similar mechanisms might take place in the collecting duct [46]. To challenge this model in M1 cells, we used cyclopiazonic acid (CPA), a well-known inhibitor of SERCA pump activity, which blocks Ca^2+^ sequestration into ER lumen and depletes the ER Ca^2+^ content. Pretreatment with 40 μM CPA for 20 min in the absence of extracellular Ca^2+^ is sufficient to empty the InsP3-sensitive ER Ca^2+^ store since subsequent addition of ATP did not increase the intracellular Ca^2+^ level in M1 cells (Appendix A). This protocol also abolishes the oscillatory Ca^2+^ responses elicited by both bafilomycin A1 (Figure 5A) and ML-SA1 (Figure 5B). However, Ca^2+^ oscillations induced by both lysosomal agents did not induce a net Ca^2+^ release from the ER since the response induced by CPA in their presence (see the curves and the histograms in Figure 5C–F) is not different when compared to those recorded in control conditions (Figure 5A,B). These experiments indicate that, on one hand, lysosomal Ca^2+^ signals need the ER to be filled to fuel Ca^2+^ to sustain the oscillatory response. 

On the other hand, we can safely exclude a direct effect of bafilomycin A1 and ML-SA1 on the ER Ca^2+^ content. In agreement with these observations, 2-APB, which selectively inhibits InsP3Rs in the absence of extracellular Ca^2+^, greatly reduced the number of cells responding to both bafilomycin A1 (Figure 5G, in %: bafilomycin A1 73.51 ± 4.51 vs. 2-APB + bafilomycin A1 0.98 ± 0.98, *p* = 0.0005) and ML-SA1 (Figure 5H, in %: ML-SA1 51.57 ± 2.93 vs. 2-APB + ML-SA1 8.25 ± 1.11, *p* = 0.0064), clearly indicating the presence of a functional coupling between lysosomes and ER, as previously reported [13,45]. The inhibition of lysosomal Ca^2+^ release by either CPA or 2-APB strongly suggests that tonic ER Ca^2+^ release through InsP3Rs is required to refill lysosomes with Ca^2+^ [15,45].

### 2.4. Effect of ML-SA1 and Bafilomycin A1 on AQP2 Intracellular Localization and Cyto-Skeleton Remodeling in MCD4 Cells

It is known that vasopressin-induced intracellular Ca^2+^ increase and cAMP production can promote the fusion of intracellular vesicles harboring AQP2 with the plasma membrane [37,47,48]. In the presence of vasopressin, the increase in cAMP elicited by the vasopressin receptor activates PKA which phosphorylates the AQP2 harbored in exocytic vesicles; the concomitant rise in intracellular Ca^2+^ allows the accumulation of AQP2 on the apical plasma membrane, thereby increasing membrane water permeability. Therefore, we evaluated if the activation of local lysosomal Ca^2+^ release events could modulate AQP2 accumulation at the plasma membrane of M1 cells stably expressing AQP2 (known as MCD4 cells). The panel in Figure 6 reports the confocal analysis of AQP2 localization in resting cells (CTR) and after forskolin (FK)/IBMX stimulation (FK + IBMX), a cAMP-increasing treatment that resulted in AQP2 redistribution from intracellular storage vesicles to the plasma membrane of renal principal cells. This effect was paralleled by a significant depolymerization of the actin cytoskeleton (see each inset, where phalloidin is marked in green). Strikingly, the effect of TRPML1 activation (ML-SA1) on both AQP2 membrane accumulation and depolymerization of the actin cytoskeleton was comparable to that obtained by maximal FK/IBMX stimulation. Inhibition of TRPML1 with ML-SI1, which was ineffective per se (ML-SI1), completely prevented ML-SA1-induced AQP2 plasma membrane accumulation and cytoskeleton depolymerization (ML-SA1 + ML-SI1). 

What we found even more interesting is that bafilomycin A1, which exerted similar Ca^2+^ signaling events, redistributed AQP2 in a perinuclear region without affecting phalloidin staining compared to control conditions (bafilomycin A1). The same perinuclear distribution of AQP2 under bafilomycin A1 stimulation was reported by the group of Dennis Brown in LL-CPK1 cells [49].

### 2.5. Effect of ML-SA1 and Bafilomycin A1 on the Swelling Phase of MCD4 Cells under Hypotonic Condition

AQP2 translocation at the plasma membrane should be matched by a change in the rate of M1 cell swelling measured as calcein quenching in response to hypotonicity. As shown by the representative traces (Figure 7A) and by the histogram summarizing the results obtained (Figure 7B), perfusion with forskolin reduced the swelling time phase (expressed as τ), which indicates a significant increase in the water membrane transport as the result of AQP2 accumulation at the plasma membrane. The activation of TRPML1 with ML-SA1 induced a similar reduction in the swelling time that returned to control level upon TRPML1 inhibition with ML-SI1. Stimulation with either ML-SI1 alone or bafilomycin A1 did not affect the value of τ.

### 2.6. Effect of TRPML1 Activation on cAMP/PKA and Ca^2+^/Calcineurin Pathways

All together, these pieces of evidence underline that (1) the discharge of Ca^2+^ from lysosomes is not the only determinant involved in AQP2 translocation/actin depolymerization/water reabsorption since ML-SA1 and bafilomycin A1 exert opposite effects, and (2) ML-SA1 share with a submaximal dose of forskolin a strong ability in activating both AQP2 membrane translocation and the increase in water permeability. The idea that the local TRPML1 activation (and likely lysosomal Ca^2+^ release) could significantly affect a tissue-defining cellular function of the collecting duct, such as AQP2-mediated water reabsorption, is completely unexplored. Therefore, at first, we tested whether the activation of TRPML1 could be associated with increases in the cAMP level and/or PKA activity and whether the antidiuretic effect of TRPML1 relied on PKA. Therefore, we first transfected M1 cells with Epac-S-H187, a loss of FRET cAMP probe that reports the increase in the intracellular cAMP as a reduction in the FRET emission ratio [50]. As shown in Figure 8A, when the cells were stimulated with ML-SA1, no change in the intracellular cAMP level was recorded. However, when the same cells were challenged with forskolin/IBMX, the emission ratio was significantly reduced, clearly indicating increases in cytosolic cAMP. In parallel experiments, we used the same protocol to measure the activity of PKA in M1 cells transfected with the gain-of-FRET PKA bioreporter AKAR4 (Figure 8B) [51]. Again, when the cells were stimulated with ML-SA1, no changes in the PKA level were recorded, while in response to FK/IBMX, the emission ratio increased, thereby indicating a rise in PKA activity. In addition, pretreatment with H89, a specific inhibitor of PKA, did not block either the translocation of AQP2 induced by the activation of TRPML1 in M1 cells or actin depolymerization (Figure 8C and inset), indicating that the cAMP/PKA pathway is not involved in the ML-SA1-induced change in AQP2 intracellular localization. 

We next wondered whether the increased amount of AQP2 localized at the apical membrane could be the direct result of the depolymerization of the actin cytoskeleton that we observed upon TRPML1-induced lysosomal Ca^2+^ release. It is well known that lysosomal calcium efflux through the lysosomal Ca^2+^ channel TRPML1 can activate the Transcription Factor EB (TFEB) through CaN-dependent dephosphorylation [52]. CaN is a Ca^2+^/calmodulin-dependent serine/threonine protein phosphatase that regulates cytoskeleton organization in different cell types, such as neurons, myocytes, podocytes and renal tubular cells. Of note, the control of cytoskeleton organization can be exerted by CaN in two ways: a direct control of filament organization through dephosphorylation of cytoskeletal organizing proteins (such as cofilin, WAVE-1 and synaptopodin) and a long-lasting transcriptional effect mediated by the nuclear factor of activated T-cells (NFAT).

Firstly, we evaluated if the ML-SA1-induced effect on AQP2 translocation and actin depolymerization is Ca^2+^-dependent. Pretreatment with the Ca^2+^ chelator BAPTA-AM, that did not change AQP2 intracellular distribution and the polymerization state of the actin cytoskeleton per se (Figure 9A, BAPTA), was able to completely inhibit the effect of ML-SA1 (Figure 9A, MLSA1 + BAPTA). Note that the 30 min pretreatment with 20 µM BAPTA significantly blocked the lysosomal Ca^2+^ release induced by ML-SA1 in the presence of extracellular Ca^2+^ (data not shown). In addition, cyclosporine A (CsA), which is a selective CaN inhibitor [53,54], reduced AQP2 translocation and actin depolymerization either when used alone (Figure 9B, CsA) or when used in combination with ML-SA1 (Figure 9B, ML-SA1 + CsA). These findings indicate that the antidiuretic effects of TRPML1 activation in M1 cells rely upon activation of the Ca^2+^/CaN pathway. In addition, to further address whether CaN also mediates the TRPML1-induced nuclear translocation of NFAT, M1 cells were transfected with NFAT-GFP. Then, its subcellular localization, expressed as the ratio between the number of cells expressing the probe in the nucleus (* in the Figure 9C) vs. those expressing the probe in the cytosol (# in the Figure 9C), was compared in control and treated cells (Figure 9D). Of note, 2 h exposure to ML-SA1 led to an increase in the percentage of M1 cells with NFAT-positive nuclei when compared to the control condition (see the panel in Figure 9D, ML-SA1). ML-SA1-induced translocation of NFAT-GFP was inhibited by pretreatment with 1 μM CsA (Figure 9D, ML-SA1+ CsA) or ML-SI1 **(**Figure 9D, ML-SA1+ ML-Sl1). Finally, bafilomycin A1 treatment did not affect NFAT localization (Figure 9D, bafilomycin A1).

### 2.7. TRPML1 Stimulation Promotes AQP2 Accumulation on the Apical Membrane of Collecting Duct Cells of Freshly Isolated Kidney Slices

Finally, we exploited the ex vivo stimulation of freshly isolated mouse kidney slices to assess the physiological relevance of the signaling pathway recruited downstream of TRPML1 stimulation. To physiologically stimulate AQP2 exposure on the apical membrane of the cells lining the CD, we used the V2R specific agonist dDAVP (dDAVP). As shown in Figure 10A, TRPML1 activation increased the rate of AQP2 apical exposure, similarly to dDAVP, when compared to kidney slices in control condition (CTR). Of note, pretreatment with the channel blocker largely prevented the dramatic redistribution of AQP2 on the apical membrane induced by the exposure to ML-SA1 (ML-SA1 + ML-Sl1). As showed in Figure 10B, the analysis of the extent of AQP2 fluorescent signal from the apical membrane (AM) to the nucleus (N) indicates that upon TRPML1 activation, AQP2 is preferentially localized at the apical plasma membrane rather than in cytosolic compartments, resembling the effect of dDAVP. Pretreatment with the ML-SI1 increased the extent of AQP2 fluorescent signal in the cytosolic compartment underneath the apical plasma membrane as the control condition.

## 3. Discussion

This study highlights the relevance of the Ca^2+^-releasing activity of the endo/lysosomal non-selective cation channel TRPML1 on the AQP2-mediated water handling of the collecting duct (CD) in the absence of vasopressin stimulation.

AQP2 redistribution back and forth the apical membrane of CD cells relies on the activation of a multifactorial signaling network. Once activated by AVP, V2R increases cytosolic cAMP levels which in turn activate PKA and stimulate PKA tethering to AQP2-containing vesicles, thereby leading to AQP2 phosphorylation. In addition, vasopressin triggers intracellular Ca^2+^ mobilization. Consequently, these vesicles dock and fuse with the apical membrane of principal cells, increasing AQP2 accumulation and water permeability so that the CD can significantly reabsorb water and concentrate urine. The transcription factors, cAMP-responsive binding protein (CREB) and CaN-dependent NFAT, are also associated with AQP2 homeostasis during antidiuresis [55].

Herein, we showed, for the first time, that lysosomal Ca^2+^ signaling events elicited specifically by TRPML1 activation also exert significant effects on in vitro and ex vivo AQP2 accumulation at the apical membrane of principal cells lining the CD (Figure 6 and Figure 10). ML-SA1- and bafilomycin A1-induced cytosolic Ca^2+^ oscillations, although similar, induced different effects on AQP2 membrane accumulation, polymerization of the actin cytoskeleton and AQP2-mediated water permeability (Figure 3,Figure 6 and Figure 7). We strongly believe that the experimental evidence collected in this study points to local Ca^2+^ release events through TRPML1 as important determinants for AQP2 trafficking. When we used ML-SI1, a specific inhibitor of TRPML1, we completely abrogated ML-SA1-induced Ca^2+^ oscillations and consequently AQP2 apical accumulation both in vitro and ex vivo (Figure 4, Figure 6 and Figure 10). In addition, when ML-SA1-induced lysosomal Ca^2+^ release events were buffered by the cytosolic Ca^2+^ chelator BAPTA (Figure 9), AQP2-transporting vesicles did not change their intracellular distribution even in presence of ML-SA1.

At first sight this evidence could be classified as obvious because of the large amount of data associating cytosolic Ca^2+^ increases and the regulated fusion of secretory vesicle with the plasma membrane of many cell types. As for vesicles harboring AQP2 in CD cells, rises in intracellular Ca^2+^ levels have been proved of capital importance for synthesis and trafficking of AQP2 [34,47,48,56,57]. For instance, we have previously shown that, in renal cells, the Ca^2+^-mobilizing agents, ATP and rosiglitazone, stimulate a robust mobilization of AQP2 to the apical plasma membrane [34]. In this scenario, the role played by TRP channels has attracted widespread interest from the scientific community. These channels sense and transduce a variety of extracellular cues driving Ca^2+^ influx into cells to control a wide spectrum of cellular responses. Plasma membrane expression and functional activity of TRPC3, TRPC6, TRPV4 and TRPV6 have been reported in CD cells and CD-derived cultures [34]. The activation of these channels orchestrates Ca^2+^ responses that are mainly composed by a remarkable Ca^2+^ influx often associated with an additional Ca^2+^ release by intracellular stores, such as the ER. These robust Ca^2+^ signals can rapidly invade the bulk of the cytosol thus allowing the control of molecular effectors localized throughout the cell. The Ca^2+^ signaling events associated so far to the activation of TRPML1 seem to escape this narrative. TRPML1 is a nonselective lysosomal cation channel that mediates Ca^2+^ efflux upon activation with either endogenous (PI(3,5)P2) or synthetic ligands (like ML-SA1) [26]. TRPML1-induced Ca^2+^ signaling events locally regulate various lysosomal functions, including lysosomal exocytosis, membrane trafficking and lysosomal biogenesis [58,59]. Unlike TRPML2 and TRPML3 which have been localized also at the plasma membrane, TRPML1 expression and localization is so far limited to the lysosomal membrane even though the constitutive recycling of lysosomes with the cell surface suggests the occasional presence of the channel at the plasma membrane. However, the dynamic nature of lysosomes recycling back and forth the plasma membrane complicates the experimental evaluation of TRPML1 surface expression with conventional techniques, such as immunofluorescence and biotinylation. Here, when we used ML-SA1 to activate TRPML1 in the presence of extracellular Ca^2+^, we found that intracellular Ca^2+^ level either oscillates or undergoes a persistent increase until agonist removal (Figure 3). If the Ca^2+^ oscillatory response is often reported upon lysosomal Ca^2+^ release [20,24], a rapid and persistent Ca^2+^ increase hints at the involvement of plasma membrane Ca^2+^ channels or a robust contribution by intracellular Ca^2+^ stores. As a matter of fact, when we repeated the same response in the absence of extracellular Ca^2+^, we likely isolated the lysosomal Ca^2+^ contribution that is featured by an oscillatory pattern (Figure 3). Whether these Ca^2+^ oscillations originate solely by lysosomal Ca^2+^ discharge or upon interaction with other organelles such as ER or mitochondria is still a matter of debate.

Our results support a spatial model according to which the lysosomal Ca^2+^ compartment in renal cells is likely to be refilled by the ER, as also reported in other cell types [15,39,45]. Since CPA and 2-APB completely inhibited the Ca^2+^ release events elicited by both bafilomycin A1 and ML-SA1 (Figure 5), cytosolic Ca^2+^ oscillations were initially thought to be sustained by lysosomal Ca^2+^-triggered CICR via the InsP3R, as widely reported for lysosomal Ca^2+^ signaling. However, this model does not fully explain the retained ability of CPA to release the same extent of ER Ca^2+^ after either bafilomycin A1 or ML-SA1 stimulation. Conversely, in primary cultured human fibroblasts [20], GPN-induced Ca^2+^ oscillations were depleted upon CPA pretreatment and the same finding was obtained in human cardiac mesenchymal stromal cells [24], human circulating endothelial colony forming cells [21] and human metastatic colorectal cancer cells [22]. On the other hand, Garrity and coworkers demonstrated that blocking SERCA with thapsigargin or inhibiting InsP3Rs with 2-APB depleted the lysosomal Ca^2+^ pool, thereby abrogating ML-SA1-induced intracellular Ca^2+^ release [45]. Similarly, the measurement of lysosomal Ca^2+^ concentration with selectively targeted aequorin confirmed that preventing SERCA activity with CPA inhibited lysosomal Ca^2+^ replenishment and suppressed lysosomal Ca^2+^ mobilization [13]. Therefore, our findings support a model according to which tonic ER Ca^2+^ release through InsP3Rs supports lysosomal Ca^2+^ refilling in a pH-dependent manner in M1 cells. Since ML-SA1 does not always lead to an increase in [Ca^2+^]i under free Ca^2+^ conditions, we envisage the existence of a lysosomal population which requires basal extracellular Ca^2+^ entry to be fully replenished. Whether this constitutive influx of Ca^2+^ is transferred to lysosomes directly or indirectly via the ER remains the object of future investigation.

Lysosomal Ca^2+^ release through TRPML1 in renal collecting duct cells can in turn regulate water homeostasis by recruiting CaN. This Ca^2+^/calmodulin dependent serine/threonine protein phosphatase is crucial for renal function, as suggested by the nephrotoxicity associated with the use of its blocker CsA. Of note, CaN can affect cytoskeleton organization in the kidney acting in two ways: a direct control of filament organization through dephosphorylation of cytoskeletal organizing proteins (such as cofilin, WAVE-1 and synaptopodin) and a long-lasting transcriptional effect mediated by NFAT [60]. For instance, we showed that a brief exposure to CsA can reliably block ML-SA1-induced modulation of both AQP2 mobilization and actin remodeling in collecting duct cells (Figure 9). However, prolonged stimulation with ML-SA1 is also able to induce NFAT nuclear translocation, which is largely employed as a proxy for CaN activation and is likely to exert a long-term transcriptional control on water homeostasis. In this context, our study is in line with the findings reported in mpkCCD_c14_ murine collecting duct principal cells, where AQP2 expression is regulated by the Ca^2+^/CaN-NFAT pathway in response to intracellular Ca^2+^ signals evoked by hypertonicity and other extracellular stimuli [55].

Ca^2+^ signaling events are significantly influenced by intracellular pH changes [42,61]. Nonetheless, the fact that TRPML1 activation does not affect either cytosolic or lysosomal pH (Figure 3) facilitates our understanding of the molecular mechanisms downstream lysosomal Ca^2+^ release. The weak base GPN was firstly used as a lysosomotropic agent to evaluate lysosomal Ca^2+^ storage and signaling events in renal cells [39]. However, the use of GPN to evoke Ca^2+^ events has been limited in this study because it alkalizes the intracellular pH potentially leading to the onset of intracellular Ca^2+^ responses completely unrelated to lysosomes, as already demonstrated in primary cultured human fibroblasts [42]. In addition, vesicle acidification is pivotal in both the constitutive and regulated recycling of AQP2 both in the presence and absence of vasopressin stimulation [49]. In this study, we have shown that bafilomycin A1 exposure in renal cells elicited lysosomal oscillatory Ca^2+^ transients likely associated with lysosomal matrix alkalization (Figure 3). Under this experimental condition, despite the similar Ca^2+^ release events induced by TRPML1 activation, bafilomycin A1 blocks the recycling pathway of AQP2 in a perinuclear compartment (Figure 6). This finding confirms the elegant demonstration provided by Dennis Brown 20 years ago that bafilomycin A1 blocks AQP2 translocation in a perinuclear location that is adjacent to the Golgi [49]. Thus, lysosomal Ca^2+^ release and the intraluminal pH of the endo-lysosomal system seem to strongly participate in controlling the fate of AQP2-transporting vesicles within renal cells. 

Considering that the effect induced by ML-SA1 on AQP2 trafficking/water transport is as large as that induced by submaximal doses of the cAMP-mediated agonists FK/IBMX, we wondered whether TRPML1 activation could recruit the cAMP/PKA pathway. However, we demonstrated that all the effects mediated by TRPML1 activation are totally unrelated to cytosolic cAMP rises/PKA activation (Figure 8) since (a) cAMP intracellular levels, as measured with a loss-of-FRET probe, are unchanged after ML-SA1 treatment; (b) ML-SA1 does not increase activation of PKA, as measured with a gain-of-FRET probe; and (c) the effect of ML-SA1 on AQP2 trafficking is not prevented by incubation with the PKA inhibitor H89.

Collectively, these results point to local lysosomal Ca^2+^ release events mediated by TRPML1 as a new player in the modulation of AQP2-mediated water homeostasis in the absence of AVP stimulation (Figure 11). It is important to mention that lysosomal Ca^2+^ signaling events through TRPML1 are also impaired in lysosomal storage disorders which often manifest renal phenotypes related to impaired water homeostasis such as polyuria and reduced urine osmolality [9]. Therefore, the complete understanding of the mechanisms through which TRPML1 mediates AQP2 water homeostasis could be of pivotal importance in the attempt to correct renal defects in LSD.

## 4. Materials and Methods

### 4.1. Reagents and Antibodies

2-aminoethyl diphenylborinate (2-APB), (cat no. D9754), ammonium chloride (cat no. 213330), adenosine 5′-triphosphate (ATP, cat no.A2383), cyclopiazonic acid (cat no. C1530), cyclosporin A (cat no. C3662), 1-deamino-8-arginine vasopressin (dDAVP, cat no. V1005), ethylene glycol-bis(2-aminoethylether)-N,N,N′,N′-tetraacetic acid (EGTA, cat no. E3889), GW405833 (ML-SI 1, cat no. G1421), H-89 (cat no. B1427), 3-isobutyl-1-methylxanthine (IBMX, cat no. I5879), ML-SA1 (cat no. SML0627) and Nigericin (cat. # N7143) were from Merck KGaA (Darmstadt, Germany). Bafilomycin A1 (cat no. S1413) was from Selleckchem (Huston, TX, USA). BAPTA AM (cat no. B1205), BCECF AM (cat no. B1170), Fura-2 AM (cat no. F1221), Lipofectamine 2000 (cat no. 18324012), Phalloidin-488 (cat no. A12379) and Calcein-AM (cat no. C1430) were from Thermo Fisher Scientific (Waltham, MA, USA). Forskolin (cat no. FOR010b) was from Fermentek (Jerusalem, Israel). Glycyl-N-2-naphthalenyl-L-phenylalaninamide (GPN, cat no. 14634) was from Cayman (Ann Arbor, MI, USA), and the Lysosome Staining Kit was from Abnova (cat no. KA4113, Taipei, Taiwan). The rabbit affinity-purified polyclonal antibody against human AQP2 was previously described [62]. Polyclonal anti-TRPML (cat no. ACC-081) was from Alomone (Jerusalem, Israel), and recombinant anti-LAMP-1 (cat no. ab208943) was from Abcam (Cambridge, UK).

### 4.2. Cell Culture

Immortalized mouse cortical collecting duct M1 cells [63] and MCD4 cells, a clone of M1 cells stably transfected with human AQP2, were cultured as described elsewhere [64,65].

### 4.3. Evaluation of Cytosolic Ca^2+^ Levels, Intracellular pH and Lysosomes Visualization

Cells were seeded on glass coverslips (Ø 15 mm). Cytosolic Ca^2+^ evaluations were performed with the ratiometric dye Fura-2 AM and intracellular pH changes were recorded using the ratiometric dye BCECF AM. Cells were loaded with either 4 µM Fura-2 or 2.5 µM BCECF for 15–30 min at 37 °C and then rinsed with saline solution to wash out dye retained extracellularly. Coverslips with dye-loaded cells were analyzed with a set-up described in detail in previous papers of our group [35,66,67,68]. Briefly, during the experiment, a Ringer’s solution (containing 140 mM NaCl, 5 mM KCl, 1.2 mM CaCl2, 1 mM MgCl2, 5 mM glucose, 10 mM HEPES with a final pH of 7.40) was used to perfuse the cells. Ca^2+^-free experiments were carried out in a Ringer’s solution without Ca^2+^ with the addition of 50 µM EGTA. Fura-2 and BCECF ratios were acquired using a ratio imaging set-up running Metafluor software (Version 7.7.3.0, Molecular devices, San Jose, CA, USA). Each coverslip, mounted in an open-top perfusion chamber, was placed on the heated stage of a Nikon TE200 inverted microscope (Nikon, Tokyo, Japan). For Fura-2 experiments, cells were excited alternately at 340 nm and 380 nm through a 40× (numerical aperture [NA], 1.4) oil immersion objective. For BCECF experiments, cells were excited alternately at 440 nm and 490 nm. The excitation wavelengths were generated using a monochromator system in the path of a 75-W xenon light source. Pairs of fluorescence images for both dyes (emission collected at 520 nm) were captured by a cooled CCD camera CoolSNAP HQ (Photometrics, Tucson, AZ, USA) every 5 s and converted to a ratio image by the Metafluor software. We used a specific positive internal control at the end of each run. For Fura-2 experiments, cells were stimulated with a submaximal dose of ATP (100 µM), and ammonium chloride was used to induce maximal pH changes in BCECF-loaded M1 cells. To analyze Ca^2+^ membrane resting permeability, an Mn^2+^ quench experiment was performed [69]. Mn^2+^ entered cells through Ca^2+^ pathways and quenched Fura-2 fluorescence. In the presence of Mn^2+^, the quenching of Fura-2 fluorescence excited at the Ca^2+^-insensitive wavelength of 356 nm can be considered as evidence of Ca^2+^ influx. Finally, the same set-up was used to acquire images of the acidic endo-lysosomal system of M1 cells. The Lysosome Staining Kit (LSK) is a permeable hydrophobic compound that visualize the acidic compartment of live cells in orange fluorescence (Ex/Em = 575/600 nm). LSK enhances its fluorescence upon entering the lysosome, while a reduction in the lysosomal pH gradient causes a drop in the fluorescence signal. Therefore, transmitted light images of cells were used to draw the region of interest corresponding to positive staining to gain information regarding LSK fluorescence as an index of lysosomal pH acidity. The probe was loaded in M1 cells following the manufacturer’s instructions. For calculation of the slope upon treatments with lysosomal agents, the data recorded were fitted to a line using non-linear regression analysis. Slopes are reported in the result section as mean ± SEM.

Regardless of the probe used in our experiments, we decided to show traces of 5 out of the total recorded cells for each protocol. Figure legends report the number of times each protocol has been repeated (*m*) and the number of responsive/total cells (*n*). For each experiment, the fluorescence signal acquired was normalized to the basal fluorescence recorded in the absence of the stimulus. Significant differences were calculated by Student’s *t*-test for paired or unpaired data.

### 4.4. FRET-Based Measurements of cAMP/PKA in Single M1 Cells

The FRET-based sensors Epac H187 for intracellular cAMP (Addgene plasmid # 170348; http://n2t.net/addgene:170348; RRID:Addgene_170348) and AKAR4 for PKA activity (Addgene plasmid # 61619 ; http://n2t.net/addgene:61619; RRID:Addgene_61619) were gifts from Kees Jalink and Jin Zhang, respectively. Cells were seeded on Ø 15 mm glass coverslips and transiently transfected with Epac H187 or AKAR4 sensors using Lipofectamine^®^ 2000. Cells were mounted in a perfusion chamber and imaged using 40XPlan Fluor (NA 1.30) oil immersion objective lens. Ringer’s solution was used to perfuse cells during the experiment, containing 140 mM NaCl, 5 mM KCl, 1 mM MgCl2, 10 mM HEPES, 5 mM glucose and 1.0 mM CaCl2 at pH 7.4. Real-time FRET imaging experiments were performed at room temperature using a fluorescence ratio imaging system built around a Nikon Eclipse TE2000-S inverted fluorescence microscope (Nikon, Tokyo, Japan) equipped with a cooled CCD camera CoolSNAP HQ (Photometrics, Tucson, AZ, USA). MetaFluor software was used to acquire the FRET emission ratios (440 nm excitation, 535 nm/485 nm) every 5 s. The trace reported in each figure depicts the mean emission ratio ± SEM of cells in the same coverslip. The figure legend reports the number of times each protocol has been repeated (*m*) and the number of responsive/total cells (*n*).

### 4.5. Evaluation of NFAT-GFP Translocation

M1 cells were transfected with NFATc4-GFP [70,71] with Lipofectamine 2000 according to the manufacturer’s instructions. After 24 h, cells were either left untreated (CTR) or stimulated as follows: 100 nM bafilomycin A1 for 2 h, 100 µM ML-SA1 for 2 h alone or after 30 min pretreatment with either 1 µM cyclosporine A or 25 µM ML-SI1. Cells were fixed in PBS containing 4% paraformaldehyde and stained with DAPI for nuclear counter staining. Coverslips were mounted on a microscope slide and placed on the stage adapter of a Leica DM6000B fluorescence microscope. Cells were illuminated at 480 nm through a PL FLUOTAR L 40X/0.60 DRY objective, and the emitted fluorescence collected at 520 nm. The same field was also illuminated at 360 nm, and the emitted fluorescence was collected at 457 nm to visualize nuclear DAPI. Five different fields for each treatment from three independent experiment were acquired blindly. Cells expressing NFAT-GFP were counted, and ImageJ was used to keep track of the nuclei vs. cytosol-positive staining of the probe using DAPI colocalization to confirm NFAT nuclear translocation. The results were expressed as NFAT nuclear to cytosol ratio for each treatment. The figure legend reports the total number of cells analyzed for each treatment. Significant differences were calculated by ordinary one-way ANOVA and Dunnett’s multiple comparisons test.

### 4.6. Immunofluorescence and Confocal Microscopy

For the immunofluorescence experiments, MCD4 cells were cultured on glass coverslips and used after full confluence. Twenty-four hours before the experiment, cells were treated with the cyclooxigenase inhibitor indomethacin 50 μM overnight in the culture medium to prevent the increase in basal cAMP concentration due to autocrine/paracrine stimulation of P2-purinergic receptors [72,73], as previously reported [34]. M1 cells were left unstimulated (CTR) or underwent the following treatments: 100 µM FK+ 500 µM IBMX for 15 min; 100 µM ML-SA1 for 15 min; 25 µM ML-SI1/10 µM H89/20 µM BAPTA or 1 µM cyclosporine for 30 min either alone or followed by a co-incubation with ML-SA1; 100 nM bafilomycin A1 for 30 min. Cell monolayers were either fixed in cold methanol or in PBS containing 4% paraformaldehyde and permeabilized with 0.1% Triton X-100 in PBS for 10 min. Cells were blocked with 1% BSA in PBS, then were incubated overnight at 4 °C with primary antibodies. Bound antibodies were detected with Alexa Fluor secondary antibodies (Thermo Fisher Scientific, Waltham, MA, USA). Planar xy images were collected using a 63X objective (HCX PL APO CS63.0× 1.40 OIL UV) mounted on a Leica TCS-SP5 (Leica, Mannheim, Germany) confocal microscope.

### 4.7. Fluorescence-Quenching Assays

A total of 40,000 MCD4 cells were seeded in each well of a 96-well black, clear bottom Corning microplates (#3603, Corning, New York, NY, USA) and grown to 90% confluence. Cells were treated overnight with indomethacin 50 µM. The day after, cells were loaded with 10 μM calcein AM (Molecular Probes, Eugene, OR, USA) dissolved in complete medium with indomethacin for a total time of 45 min at 37 °C. After 15 min, during calcein incubation, cells were left untreated (CTR) or stimulated for the remaining 30 min as follows: 5 µM FK+ 500 µM IBMX; 100 µM ML-SA1; 25 µM ML-SI, 100 nM bafilomycin A1. When we used both the agonist and the antagonist in combination, ML-SI1 was added to the incubation mixture 20 min before ML-SA1. Cells were then rinsed in isosmotic PBS (added with 1 mM Ca^2+^ and 1 mM Mg^2+^) containing each treatment (as described above). Fluorescence signal changes in calcein-loaded glial cells after hypotonic gradient were recorded on a Flex Station3 plate reader (Molecular Devices, San Jose, CA, USA) equipped with an integrated automatic liquid handling module, as previously described [74,75]. Fluorescence was excited at 490 nm and detected at 520 nm using dual monochromators. The emitted fluorescence by calcein-loaded cells was recorded continuously for 15 s (baseline), then for 35 s, every 0.5 s, after the automated addition of an appropriate volume of distilled water to obtain 240 mOsm/L final extracellular osmolarity. Calcein fluorescence intensity increased upon the addition of the hypotonic solution due to water influx and cell swelling. Data acquisition was performed by the SoftMax Pro software (Version 5.3, Molecular Devices, San Jose, CA, USA) and the data were analyzed with Prism software (GraphPad Software, La Jolla, CA, USA). The time constant of cell swelling induced by the hypotonic stimulus was obtained by fitting the data to an exponential function. Statistically significant differences between control and treated cells exposed to a hypotonic shock were computed using one-way ANOVA analysis and Dunnett’s multiple comparisons test.

### 4.8. Kidney Slices

Sex-, age- and weight-matched C57BL/6J male mice (*N* = 6) were used for kidney slice experiments as previously reported [35,76,77]. Briefly, mice were anesthetized with isoflurane 1.5% (*v*/*v*) and killed by cervical dislocation. Kidneys were collected and thin transversal slices (250 μm) were obtained using a Mcllwain Tissue Chopper (Ted Pella Inc.; Redding, CA, USA). Slices were left at 37 °C in Dulbecco’s modified eagle/F12 medium (CTR) or stimulated for 30 min with dDAVP (100 µM) or ML-SI1 (25 µM) or ML-SA1 (100 µM), the latter given alone or after 30 min of preincubation with ML-SI1. Slices were fixed in 4% paraformaldehyde, and then treated an analyzed as previously described [78]. As primary antibody, an anti-AQP2 (1:200) and then the appropriate Alexa Fluor-conjugated secondary antibody were used. Sections were counterstained with Evans blue. The confocal images were obtained with a laser scanning fluorescence microscope Leica TSC-SP5 (HCX PL APO, ×63/1.32–0.60 oil objective). Image J (Version 1.54b) was used to calculate the fluorescent distribution of the AQP2 signal along a line drawn from the apical membrane to the nucleus. Then, the extent of the AQP2 signal within the cell was obtained measuring the distance (in µm) between the AQP2 signal on the apical plasma membrane and its disappearance in the cytosolic compartment. The figure legend reports the total number of cells analyzed for each treatment. Significant differences were calculated by ordinary one-way ANOVA and Dunnett’s multiple comparisons test.

### 4.9. Western Blotting

M1 cells were seeded on a 6-well culture plate. Cells were washed with ice-cold PBS and scraped with RIPA buffer (150 NaCl mM, 10 mM Tris, 1% Triton X-100, 0.1% SDS, 1% deoxycholate, 5 mM EDTA, pH 7.2) containing protease and phosphatase inhibitors (1 mmol/L phenylmethylsulfonyl fluoride, 10 mmol/L leupeptin, 1 mg/mL pepstatin A, 10 mmol/L NaF, 1 mmol/L sodium orthovanadate and 15 mmol/L tetrasodium pyrophosphate). Lysates were then sonicated at 60 amplitude with the microprobe sonicator Vibra-cell^®^ (Sonics and Materials Inc., Newtown, CT, USA) for 30 sec at 4 °C and insoluble material was pelleted at 13,000× *g* for 30 min at 4 °C. Then, 20 μg of supernatant was separated by standard SDS-PAGE using Mini-PROTEAN^®^ TGX Stain-Free™ Precast Gels Bio-Rad. The immunoreactive bands were detected with a ChemiDoc™ System (Bio-Rad, Hercules, CA; USA).

### 4.10. Statistical Analysis

GraphPad Prism 8 was used for statistical analysis. Whenever needed, statistical analysis was performed using Student’s *t*-test for unpaired data (Fura-2 and LSK experiments) or paired data (BCECF experiments). We conducted ordinary one-way ANOVA followed by the Dunnett’s multiple comparison test to compare stimulated vs. control condition depending on the dataset (fluorescence microscopy analysis and water permeability assay).

## 5. Conclusions

Collectively, our results point to local lysosomal Ca^2+^ signaling events mediated by TRPML1 as a new actor in the regulation of AQP2-mediated water homeostasis.

## Figures and Tables

**Figure 1 ijms-24-01647-f001:**
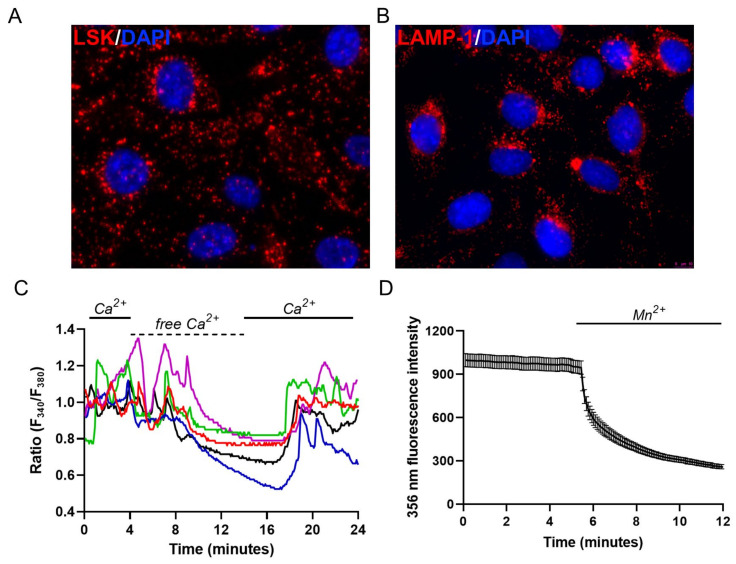
Endo-lysosomal system distribution and Ca^2+^ signaling events in M1 cells. (**A**) Representative merged epifluorescence acquisition of live M1 cells loaded with LSK (red) to stain acidic organelles and stained with DAPI (blue) to mark the nucleus; (**B**) Representative epifluorescence image of the lysosomal marker LAMP-1 (red) merged with DAPI (blue); (**C**) Ca^2+^ oscillations of 5 representative Fura-2-loaded M1 cells (*n* = 102/102 responsive cells, *m* = 3 experiments) in the presence (Ca^2+^ 1.2 mM, full lines) or in the absence of extracellular Ca^2+^ (free Ca^2+^, dotted line). Each color line represents an individual cell; (**D**) Quenching of Fura-2 after addition of 0.5 mM Mn^2+^ in the absence of extracellular Ca^2+^ at its isosbestic point. The reported values are means ± SEM from all the responsive cells on a single coverslip (*n* = 123/123 responsive cells, *m* = 3 experiments).

**Figure 2 ijms-24-01647-f002:**
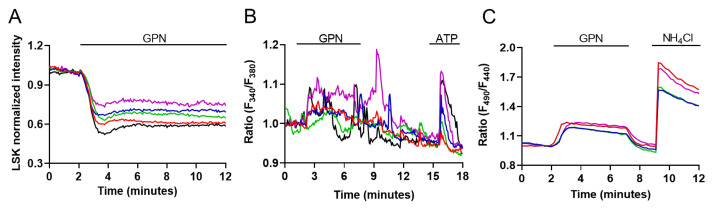
GPN evokes lysosomal Ca^2+^ signaling events and cytosolic alkalization in M1 cells. (**A**) Effect of the lysosomotropic agent GPN (200 µM) on the endo-lysosomal accumulation of LSK in M1 cells (*n* = 89/89 responsive cells, *m* = 4 experiments); (**B**) Measurements of cytosolic Ca^2+^ in response to both 200 µM GPN and 100 µM ATP in Fura-2-loaded M1 cells (*n* = 47/69 responsive cells, *m* = 3 experiments); (**C**) Cytosolic pH measurement in response to 200 µM GPN and 30 mM NH_4_Cl used as internal positive control in BCECF-loaded M1 cells (*n* = 91/91 responsive cells, *m* = 3 experiments). All graphs show 5 representative traces for each experiment. Each color line represents an individual cell.

**Figure 3 ijms-24-01647-f003:**
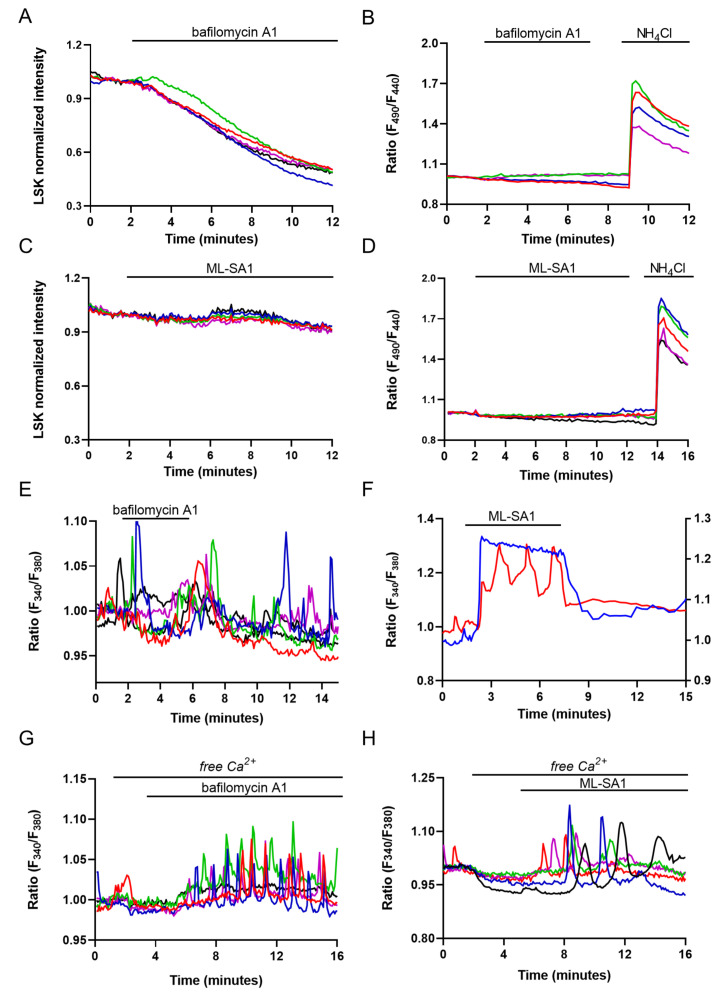
Bafilomycin A1 and ML-SA1 do not change cytosolic pH and induce similar effects on intracellular Ca^2+^ concentration in M1 cells. (**A**) Effect of 100 nM bafilomycin A1 on lysosomal fluorescence upon staining with LSK (*n* = 98/98 responsive cells, *m* = 4 experiments) and (**B**) cytosolic pH as compared to 30 mM NH_4_Cl (*n* = 0/150 responsive cells, *m* = 4 experiments). The two graphs show 5 representative traces for each experiment. Each color line represents an individual cell; (**C**) Effect of 100 µM ML-SA1 on lysosomal fluorescence (*n* = 0/107 responsive cells, *m* = 4 experiments) and (**D**) cytosolic pH as compared to 30 mM NH_4_Cl (*n* = 0/104 responsive cells, *m* = 3 experiments). The two graphs show 5 representative traces for each experiment. Each color line represents an individual cell; (**E**) Ca^2+^ oscillations evoked by 100 nM bafilomycin A1 in M1 cells loaded with Fura-2 (*n* = 19/83 responsive cells, *m* = 3 experiments). The graph shows 5 representative traces. Each color line represents an individual cell; (**F**) The graph shows 2 representative traces of the different Ca^2+^ responses induced by 100 µM ML-SA1 in M1 cells of the same coverslip loaded with Fura-2 (*n* = 101/124 responsive cells, *m* = 3 experiments): an oscillatory response (red trace, *n* = 26/101 responsive cells, *m* = 3 experiments) and a persistent increase (blue trace, n = 75/101 responsive cells, *m* = 3 experiments); (**G**) Effect of either 100 nM bafilomycin A1 (*n* = 80/111 responsive cells, *m* = 3 experiments) or (**H**) 100 µM ML-SA1 (*n* = 48/92 responsive cells, *m* = 3 experiments) on Fura-2-loaded M1 cells in the absence of extracellular Ca^2+^. The two graphs show 5 representative traces for each experiment. Each color line represents an individual cell.

**Figure 4 ijms-24-01647-f004:**
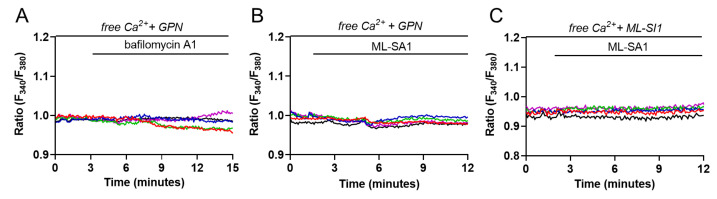
Lysosomal Ca^2+^ responses depend on lysosomal Ca^2+^ content and TRPML1 activity. (**A**) Cytosolic Ca^2+^ measurements after 30 min pretreatment with 200 µM GPN following the addition of either 100 nM bafilomycin A1 (*n* = 5/97 responsive cells, *m* = 3 experiments) or (**B**) 100 µM ML-SA1 (*n* = 3/119 responsive cells, *m* = 3 experiments) in the absence of extracellular Ca^2+^ in Fura-2-loaded M1 cells; (**C**) Cytosolic Ca^2+^ measurements of Fura-2-loaded M1 cells after 30 min pretreatment with 25 µM ML-SI1 followed by the addition of 100 µM ML-SA1 in the absence of extracellular Ca^2+^ (*n* = 3/88 responsive cells, *m* = 3 experiments). All graphs show 5 representative traces for each experiment. Each color line represents an individual cell.

**Figure 5 ijms-24-01647-f005:**
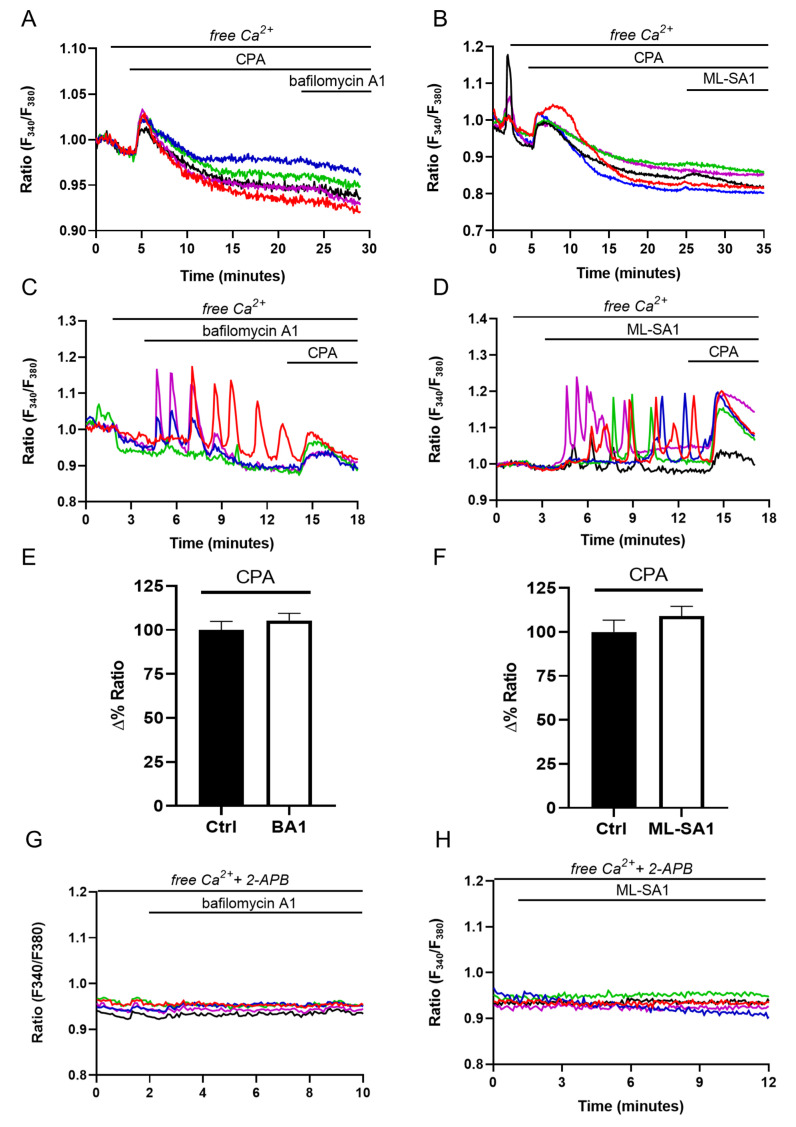
Lysosomal Ca^2+^ responses are dependent on ER Ca^2+^ content and sustained by the activity of the InsP3R. (**A**) Ca^2+^ response to the addition of either 100 nM bafilomycin A1 (*n* = 0/102 responsive cells, *m* = 3 experiments) or (**B**) 100 µM ML-SA1 (*n* = 4/166 responsive cells, *m* = 4 experiments) after 20 min pretreatment with 40 µM CPA in the absence of extracellular Ca^2+^ in Fura-2-loaded M1 cells.; (**C**) Ca^2+^ oscillations induced by either 100 nM bafilomycin A1 (*n* = 63/94 responsive cells, *m* = 3 experiments) or (**D**) 100 µM ML-SA1 (*n* = 45/94 responsive cells, *m* = 3 experiments) followed by 40 µM CPA-induced ER Ca^2+^ release in the absence of extracellular Ca^2+^. Graphs show 5 representative traces for each experiment. Each color line represents an individual cell; (**E**) Histograms summarizing the response induced by CPA alone (ctrl) or upon 10 min treatment with either 100 nM bafilomycin A1 (Δ% ratio: 100 ± 4.80, *n* = 135 cells vs. 105.4 ± 4.08, *n* = 91 cells, *p* = n.s.) or (**F**) 100 µM ML-SA1 (Δ% ratio: 100 ± 6.70, *n* = 87 cells vs. 109.1 ± 5.53, *n* = 90 cells, *p* = n.s.). Histograms show mean value ± SEM of all responsive cells. *t*-test for unpaired data was used for statistical analysis; (**G**) Ca^2+^ response to the addition of either 100 nM bafilomycin A1 (*n* = 1/112 responsive cells, *m* = 3 experiments) or (**H**) 100 µM ML-SA1 (*n* = 13/140 responsive cells, *m* = 4 experiments) after 30 min pretreatment with 40 µM CPA in the absence of extracellular Ca^2+^ in Fura-2-loaded cells. Graphs show 5 representative traces for each experiment. Each color line represents an individual cell.

**Figure 6 ijms-24-01647-f006:**
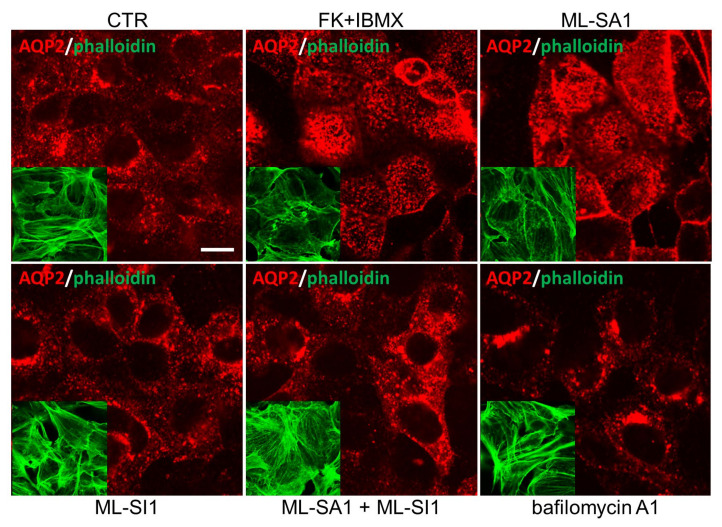
ML-SA1 and bafilomycin A1 differentially modulate AQP2 intracellular localization and cytoskeleton remodeling in MCD4 cells. M1 cells were left unstimulated (CTR) or underwent the following treatments: 100 µM FK+ 500 µM IBMX for 15 min; 100 µM ML-SA1 for 15 min; 25 µM ML-SI1 for 30 min either alone or followed by a co-incubation with ML-SA1; 100 nM bafilomycin A1 for 30 min. Cells were stained with an antibody against AQP2 (red) and phalloidin Alexafluor-488 to visualize F-actin (green, inset). Images are representative of 3 independent experiments. Scale bar: 20 μm.

**Figure 7 ijms-24-01647-f007:**
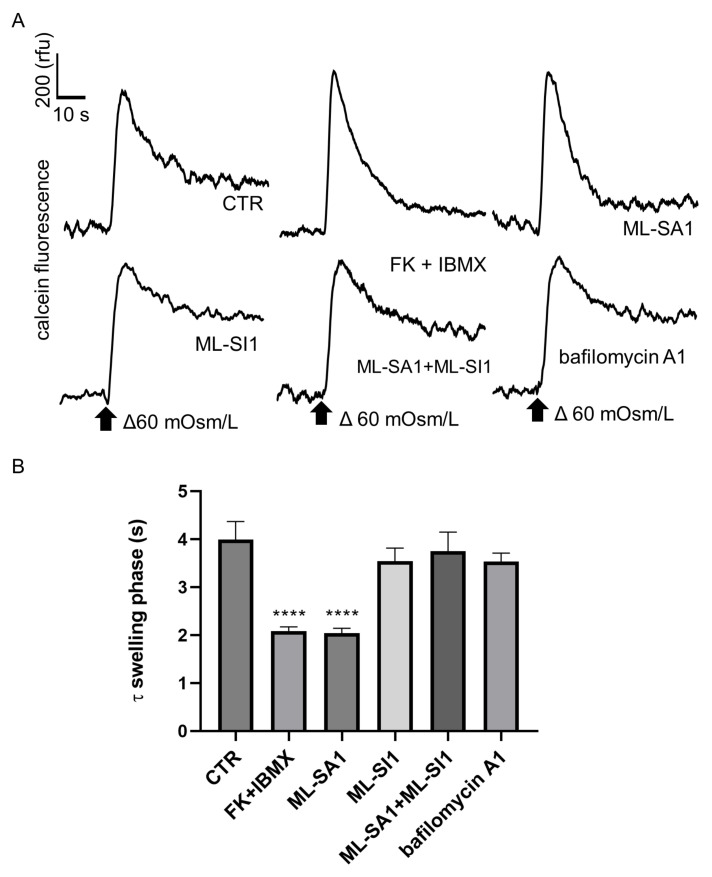
ML-SA1 and bafilomycin A1 differentially modulate the swelling phase of MCD4 cells under hypotonic condition. (**A**) Representative time courses of cell swelling phases recorded from calcein-loaded cells left untreated or stimulated as follows: 100 µM FK+ 500 µM IBMX for 15 min; 100 µM ML-SA1 for 15 min; 25 µM ML-SI1 for 30 min either alone or followed by a co-incubation with ML-SA1; 100 nM bafilomycin A1 for 30 min. The time course shows changes in fluorescence (expressed in arbitrary units, rfu) over time (s) elicited by hypotonic stimulation (Δ 60 mOsm/L) as indicated. The arrows indicate the switch in the external osmolarity; (**B**) Quantitative analysis of the cell swelling time constants (τ) in MCD4 cells. Data were obtained from 3 independent experiments. Significant differences in the means were calculated by ordinary one-way ANOVA and Dunnett’s multiple comparisons test. **** *p* < 0.0001.

**Figure 8 ijms-24-01647-f008:**
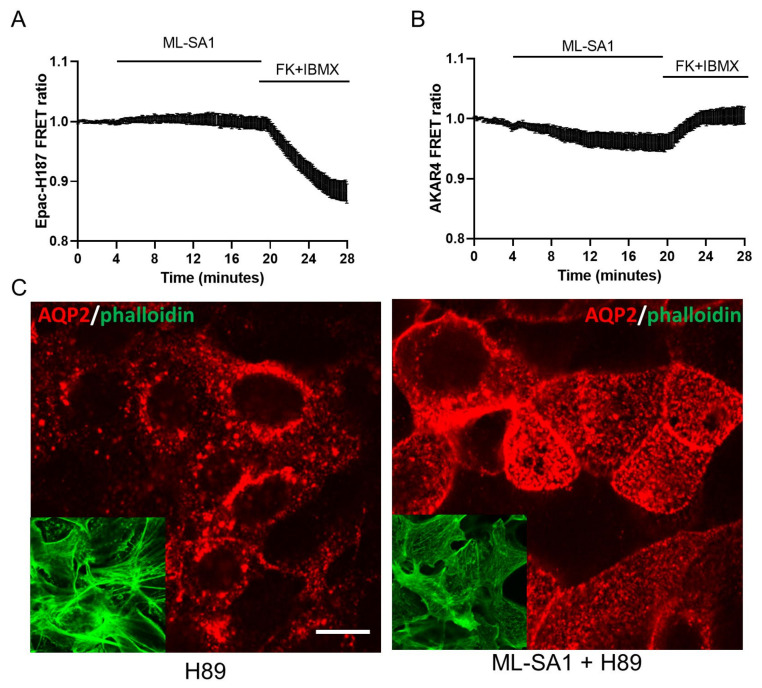
cAMP/PKA pathway is not involved in ML-SA1-induced AQP2 accumulation on the apical membrane of MCD4 cells. (**A**) M1 cells either expressing Epac-H187 (*n* = 0/15 responsive cells, *m* = 4 experiments) or (**B**) AKAR4 (*n* = 0/16 responsive cells, *m* = 4 experiments) were stimulated with 100 µM ML-SA1 for 15 min. Perfusion with 5 µM FK+ 500 µM IBMX was used as positive control for cAMP production and PKA activity, respectively. Graphs show the average ± SEM of cells on a single coverslip, at least 3 independent experiments were performed; (**C**) Cells were treated for 30 min with 10 µM H89 alone or followed by 100 µM ML-SA1 for 15 min and stained with an antibody against AQP2 (red) and phalloidin Alexafluor-488 to visualize F-actin (green, inset). Images are representative of 3 independent experiments. Scale bar: 20 μm.

**Figure 9 ijms-24-01647-f009:**
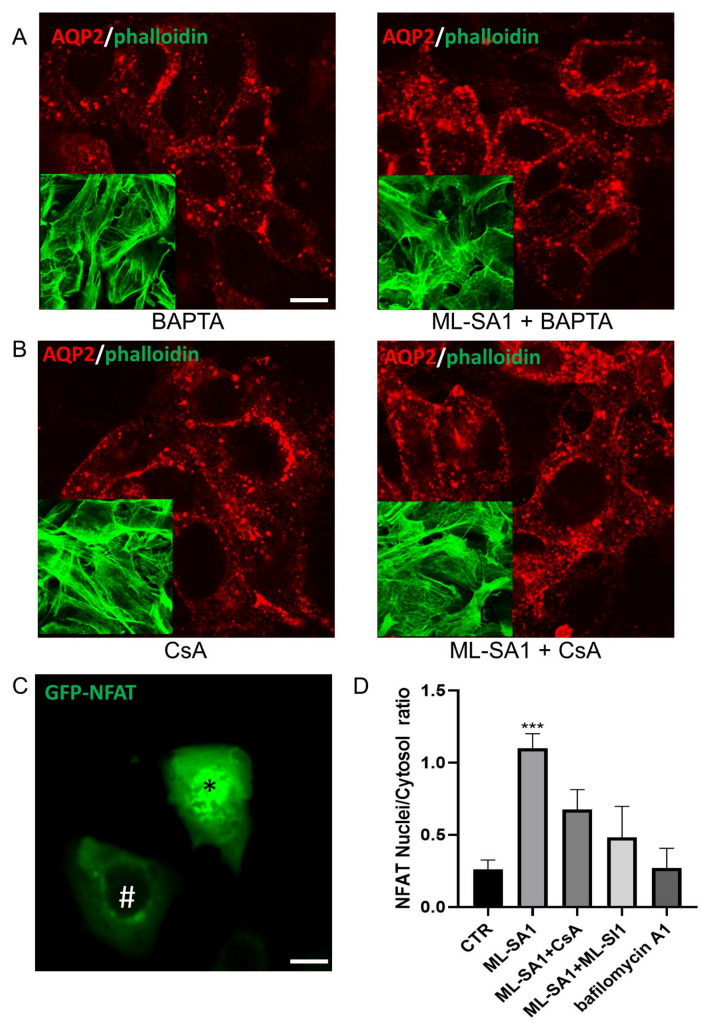
Ca^2+^/calcineurin pathway activation is involved in ML-SA1-induced AQP2 apical membrane accumulation in MCD4 cells. (**A**) Cells were treated for 30 min with 20 µM BAPTA alone or followed by 100 µM ML-SA1 for 15 min and stained with an antibody against AQP2 (red) and phalloidin Alexafluor-488 to visualize F-actin (green, inset). Images are representative of 3 independent experiments; (**B**) Cells were treated for 30 min with 1 µM CsA alone or followed by 100 µM ML-SA1 for 15 min and stained with an antibody against AQP2 (red) and phalloidin Alexafluor-488 to visualize F-actin (green, inset). Images are representative of 3 independent experiments; (**C**) Representative image of M1 cells expressing NFATc4-GFP either in the cytosol (*) or in the nucleus (#). Scale bar: 20 μm; (**D**) Histogram summarizing the ratio between the number of cells expressing NFATc4-GFP in the nucleus vs. cells expressing the probe in the cytosol. Cells were either left untreated (CTR) or stimulated as follows: 100 µM ML-SA1 for 2 h alone or after 30 min pretreatment with either 1 µM CsA or 25 µM ML-SI1;100 nM bafilomycin A1 for 2 h. Data were obtained from 3 independent experiments. Five different fields for each treatment were blindly acquired and analyzed for every independent experiment. The number of cells analyzed for each treatment is as follows: CTR, *n* = 126 cells, ML-SA1, *n* = 93 cells, ML-SA1 + CsA, *n* = 111 cells, ML-SA1 + ML-SI1, *n* = 117 cells, bafilomycin A1, *n* = 129 cells. The graph shows mean ± SEM and significant differences were calculated by ordinary one-way ANOVA and Dunnett’s multiple comparisons test. *** *p* = 0.0006.

**Figure 10 ijms-24-01647-f010:**
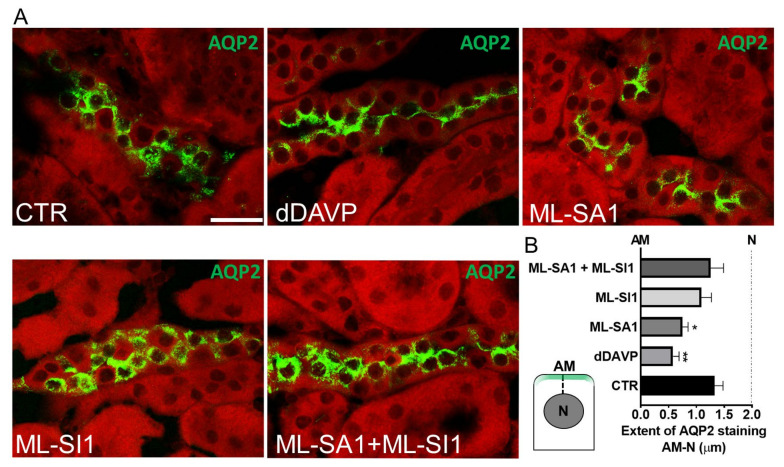
TRPML1 stimulation promotes AQP2 accumulation on the apical membrane of collecting duct cells of freshly isolated kidney slices. (**A**) Freshly isolated kidney slices from 6 wt mice were left untreated (CTR, complete medium) or stimulated for 45 min in complete medium as follows: 100 µM dDAVP; 100 µM ML-SA1 alone or after 30 min of preincubation with 25 µM ML-SI1. Confocal images of kidney sections stained with an antibody against AQP2 (green) and counterstained with Evans blue (red). The images are representative of three independent experiments. Scale bar: 10 μm; (**B**) A schematic representation of the approach used to quantify the extent of AQP2 fluorescent signal (green region) along a dotted line drawn from the apical membrane (AM) to the nucleus (N). The histogram summarizes the effects of treatments on the extent of AQP2 signal from the AM to N. For each experimental condition a total of *n* = 300 cells were blindly analyzed (*n* = 50 cells for each mouse). The graph shows mean ± SEM, and significant differences were calculated with respect to the control condition by ordinary one-way ANOVA and Dunnett’s multiple comparisons test. ** *p* < 0.01, * *p* < 0.05.

**Figure 11 ijms-24-01647-f011:**
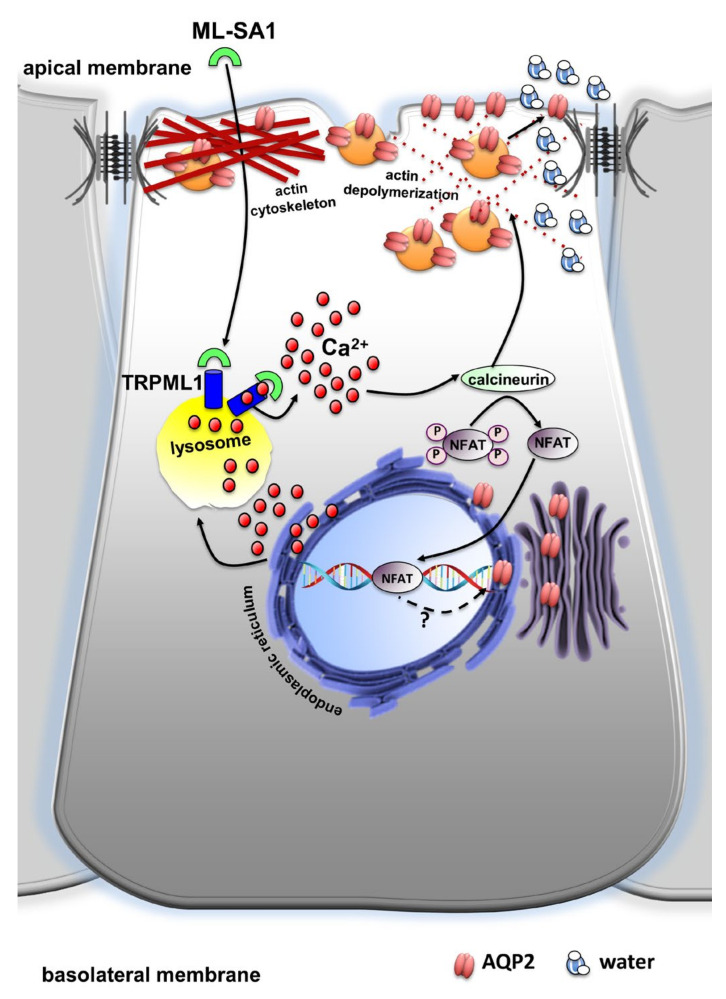
Proposed schematic diagram illustrating the effect of TRPML1 activation on AQP2-mediated water homeostasis in mouse renal collecting duct cells. ML-SA1 activates TRPML1 generating Ca^2+^ signaling events that are sustained by the endoplasmic reticulum Ca^2+^ content. The subsequent activation of the Ca^2+^/calcineurin/NFAT pathway leads to depolymerization of the actin cytoskeleton, thereby facilitating AQP2 accumulation at the apical plasma membrane and increasing water permeability. The possible role of TRPML1-induced NFAT transcriptional activity on AQP2 expression needs further investigation (question mark).

## Data Availability

Not applicable.

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
