# Peer review of "TRPML1-Induced Lysosomal Ca2+ Signals Activate AQP2 Translocation and Water Flux in Renal Collecting Duct Cells"

_ijms, 2023, doi:10.3390/ijms24021647_

Round 1
Reviewer 1 Report
The research paper titled "TRPML1-induced lysosomal Ca2+ signals activate AQP2 translocation and water flux in renal collecting duct cells" provides valuable insights into the role of TRPML1 and lysosomal Ca2+ signals in the regulation of AQP2 translocation and water flux in renal collecting duct cells. The study utilizes various techniques such as immunofluorescence, calcium imaging techniques to investigate the mechanisms underlying TRPML1-induced AQP2 translocation and water flux in renal collecting duct cells. However, the figures in this paper are not well presented, and the authors should add statistical analysis instead of just show several figures and representative traces. Additionally, the authors should provide more detailed descriptions such as the sample size and any relevant experimental conditions. This would make the results in the paper more solid and allow readers to better understand and interpret the results presented in the figures.
Major concern:
In most of the figures showed in this paper, statistical data, details of data and data analyzing are lacking, therefor, details of the methodology, such as the n number and rigorous statistics should be added for every figure, otherwise, it’s not a critical research paper.
Minor concern:
1. Figure1. C, the way the authors present the figure is very confusing, it’s difficult for the readers to find the responses from the Ca2+ containing solution.
2. Figure3. F, it seems the perfusion time of ML-SA1 isn’t right.
Author Response
We would like to thank the reviewer for the accurate analysis of our manuscript and for the constructive comments. We did our best to address all the concerns.
Italics is used for referees’ original statements.
The research paper titled "TRPML1-induced lysosomal Ca2+ signals activate AQP2 translocation and water flux in renal collecting duct cells" provides valuable insights into the role of TRPML1 and lysosomal Ca2+ signals in the regulation of AQP2 translocation and water flux in renal collecting duct cells. The study utilizes various techniques such as immunofluorescence, calcium imaging techniques to investigate the mechanisms underlying TRPML1-induced AQP2 translocation and water flux in renal collecting duct cells. However, the figures in this paper are not well presented, and the authors should add statistical analysis instead of just show several figures and representative traces. Additionally, the authors should provide more detailed descriptions such as the sample size and any relevant experimental conditions. This would make the results in the paper more solid and allow readers to better understand and interpret the results presented in the figures.
In most of the figures showed in this paper, statistical data, details of data and data analyzing are lacking, therefor, details of the methodology, such as the n number and rigorous statistics should be added for every figure, otherwise, it’s not a critical research paper.
We thank the referee for this very important suggestion. In the new version of the manuscript, we provide the number of responsive/total cells for each protocol in the figure legends. Statistical analysis has been performed whenever needed. Statistical differences have been added in the result section while information regarding the statistical analysis have been provided in the method section.
In addition, as reported in the open review section, the text of the manuscript has been significantly edited for English language and style.
Figure1. C, the way the authors present the figure is very confusing, it’s difficult for the readers to find the responses from the Ca2+ containing solution.
The referee is right. The figure 1C has been updated following the reviewer criticism.
Figure3. F, it seems the perfusion time of ML-SA1 isn’t right.
The referee is right and a new figure 3F with the proper perfusion time for ML-SA1 has been added to the manuscript.

Reviewer 2 Report
Scorza and Gerbino have written an interesting manuscript “TRPML1-induced lysosomal Ca2+ signals activate AQP2 translocation and water flux in renal collecting duct cells”, where authors characterized an important role of TRPML1 in aquaporin 2 translocation and cell swelling.
The article is well written, it has a great quality of the data presentation with sufficiently documented methods section. Scorza et al. work expands our knowledge on the possible downstream pathways affected by TRPML1 channel. This study has a high significance to the TRP field and will attract a broad interest of general audience.
The strength of the article is that it has determined that Ca2+ release through TRPML1 induced by selective agonist ML-SA1 from lysosomes triggers AQP2 translocation, actin depolymerization and cell swelling. Moreover, Ca2+ changes by TRPML1 differ from bafilomycin A1-induced pathway and are cAMP/PKA pathway independent.
I have a few suggestion to finalize this exciting research:
1) Authors presented NFAT translocation in statistical in Fig. 9D but the Fig. 10 where the accumulation of AQP2 at the PM is shown has no actual analysis of the images. I would suggest to evaluate the AQP2 translocation for the Fig. 10 for easier comprehension of the figure.
2) Methods are sufficiently describes except “4.2 Cell culture” where only citations are given for the primary cell isolation. It would be helpful to describe in brief the isolation procedure and culturing conditions.
3) For the first time the TRPML1 role in renal collecting duct cells was traced in such details. Therefore, the schematic representation of the processes described in the paper would improve the quality of the presented data.
Minor comments:
1) The Fig. 10 has no explanation how AQP2 was detected (using specific antibodies?) as it was explained in Fig. 6, 8 and 9 (“…Cells were stained with an antibody against AQP2 (red)…”).
2) Some polishing of the text is required. For example line 298: “What we found even more interesting is that bafilomycin, who exerted similar …” – “who” should be changed to “which”.
Conclusion: I greatly enjoyed reading the MS and I would enjoy reading it in print. Hence I express my strong support for publication after a small polishing of the article.
Author Response
We would like to thank the reviewer for the accurate analysis of our paper and for the constructive criticisms suitable for improving the manuscript. We did our best to address all the concerns.
Italics is used for referees’ original statements.
Scorza and Gerbino have written an interesting manuscript “TRPML1-induced lysosomal Ca2+ signals activate AQP2 translocation and water flux in renal collecting duct cells”, where authors characterized an important role of TRPML1 in aquaporin 2 translocation and cell swelling. The article is well written, it has a great quality of the data presentation with sufficiently documented methods section. Scorza et al. work expands our knowledge on the possible downstream pathways affected by TRPML1 channel. This study has a high significance to the TRP field and will attract a broad interest of general audience. The strength of the article is that it has determined that Ca2+ release through TRPML1 induced by selective agonist ML-SA1 from lysosomes triggers AQP2 translocation, actin depolymerization and cell swelling. Moreover, Ca2+ changes by TRPML1 differ from bafilomycin A1-induced pathway and are cAMP/PKA pathway independent.
We would like to thank the reviewer for his/her positive feedback. We are pleased that the reviewer finds the research important for the scientific community.
Authors presented NFAT translocation in statistical in Fig. 9D but the Fig. 10 where the accumulation of AQP2 at the PM is shown has no actual analysis of the images. I would suggest to evaluate the AQP2 translocation for the Fig. 10 for easier comprehension of the figure.
The referee is right. A new Figure 10 B has been added to the manuscript following the reviewer’s criticism. Methods and results have been modified accordingly.
Methods are sufficiently describes except “4.2 Cell culture” where only citations are given for the primary cell isolation. It would be helpful to describe in brief the isolation procedure and culturing conditions.
Thanks to the referee’s comment we realized that we did not adequately clarify that M1/MCD4 cells are immortalized mouse cortical collecting duct cells. This is now specifically stated in the method section.
For the first time the TRPML1 role in renal collecting duct cells was traced in such details. Therefore, the schematic representation of the processes described in the paper would improve the quality of the presented data.
We agree with the referee. A schematic representation of the proposed mechanism (new figure 11) has been added to the revised version of the manuscript.
Minor comments:
The Fig. 10 has no explanation how AQP2 was detected (using specific antibodies?) as it was explained in Fig. 6, 8 and 9 (“…Cells were stained with an antibody against AQP2 (red)…”).
The figure legend of Figure 10 has been changed according to the referee’s suggestion.
Some polishing of the text is required. For example line 298: “What we found even more interesting is that bafilomycin, who exerted similar …” – “who” should be changed to “which”.
According to both referee’s comments, the text underwent significant editing for English language.

Round 2
Reviewer 1 Report
The author has revised the manuscripts and answered the questions point by point